# Antioxidants Discovery for Differentiation of Monofloral Stingless Bee Honeys Using Ambient Mass Spectrometry and Metabolomics Approaches

**DOI:** 10.3390/foods12122404

**Published:** 2023-06-18

**Authors:** Wei Chean Chuah, Huei Hong Lee, Daniel H. J. Ng, Ai Ling Ho, Mohd Rosni Sulaiman, Fook Yee Chye

**Affiliations:** 1Food Safety and Security Research Group, Faculty of Food Science and Nutrition, Universiti Malaysia Sabah, Jalan UMS, Kota Kinabalu 88400, Sabah, Malaysia; 2International Food and Water Research Centre, Waters Pacific Pte Ltd., Singapore Science Park II, Singapore 117528, Singapore

**Keywords:** antioxidant, stingless bee honey, ambient MS, LC-HRMS, chemometrics, metabolomics, monofloral honey

## Abstract

Stingless bee honey (SBH) is a natural, sweet product produced by stingless bees (Meliponini tribe) that has been used as a traditional medicine to treat various illnesses. It has been shown that SBH has high nutritional value and health-promoting properties due to the presence of plant bioactive compounds from different botanical flora of the foraged nectar. In this study, the antioxidant activities of seven monofloral honeys from acacia, agarwood, coconut, dwarf mountain pine (DMP), Mexican creeper (MC), rubber, and starfruit botanical origins were investigated. The antioxidant properties of SBH studied had a range from 19.7 to 31.4 mM TE/mg for DPPH assays, 16.1 to 29.9 mM TE/mg for ABTS assays, 69.0 to 167.6 mM TE/mg for ORAC assays, and 45.5 to 89.3 mM Fe^2+^/mg for FRAP assays. Acacia honey showed the highest level of antioxidant properties. The models built from mass spectral fingerprints from direct ambient mass spectrometry showed distinct clusters of SBH by botanical origin and correlated with the antioxidant properties. An untargeted liquid chromatography-mass spectrometry (LC-MS) metabolomics approach was undertaken to identify the antioxidant compounds that could explain the unique antioxidant and compositional profiles of the monofloral SBH by its botanical origin. The antioxidants that were identified predominantly consisted of alkaloids and flavonoids. Flavonoid derivatives, which are potent antioxidants, were found to be key markers of acacia honey. This work provides the fundamental basis for the identification of potential antioxidant markers in SBH associated with the botanical origin of the foraged nectar.

## 1. Introduction

Stingless bee honey (SBH) is a natural, sweet product produced by stingless bees (Meliponini tribe) which is collected through nectar or sap collection from vegetation-rich environments and chemically modified with the aid of specific organic substances such as salivary secretions or bee enzymes, deposited, dehydrated, and stored in colonies to mature [1]. Although honey produced by honeybees (*Apis Mellifera*, tribe *Apini*) dominates the current world trade market, demand for SBH is growing because of its high nutritional values and health-promoting properties [2]. SBH has traditionally been used as medicine in sub-tropical and tropical regions such as South-East Asia, Africa, and Central and South America. It is thought that SBH has potent effects in treating many illnesses, such as respiratory infection, wound healing, gastrointestinal disorders, post-birth recovery, eye disease, asthma, and fatigue [2,3]. 

Many past studies have been devoted to the physicochemical characteristics and biological activities of SBH [4,5,6,7] due to the differences in chemical composition. Additionally, the nutritional value and health-promoting properties of SBH have been identified [2]. It has been reported that SBH is an excellent natural antibacterial against a wide range of pathogens, including antibiotic-resistant strains [8,9,10,11,12,13]. Several studies have shown that SBH could be used as a wound healing agent as it has a high scavenging capacity against reactive oxygen species and improved dermal fibroblast proliferation [14,15]. In vitro studies have reported the chemo-preventative properties of SBH against colon cancer cells and antidiabetic properties through the regulation of enzymes α-amylase and α-glucosidase [16,17]. Furthermore, there is a growing wealth of evidence for the medicinal properties of SBH and its importance to food and nutrition.

Bioactive substances such as polyphenols, vitamins, Maillard reaction products, carotenoid-like substances, organic acids, amino acids, and minerals contribute to the beneficial properties of honeys [18,19]. Entomological differences have been reported in some studies where the antioxidant activities of SBH were found to be stronger than honeybee honey [20,21,22,23]. Botanical origin is another predominant factor influencing bioactive compositions, in which antimicrobial activity was reported to be higher in specific botanical sources [24,25,26,27]. Margaoan et al. [28] reported increasing interest in monofloral honey for medicinal purposes due to the presence of phytochemicals linked to health benefits. The composition of these compounds in honeys can be influenced by many variables such as botanical source, entomology, geographical location, environment condition, harvesting season, processing technique, and storage condition [29,30]. The integration of chemical analysis followed by chemometric analysis using various clustering and classification algorithms has been successfully used to identify honey origin [31]. Unfortunately, the complexity and inconsistency of these variables make it difficult to pinpoint the exact conditions conferring the unique medicinal properties of SBH. Therefore, there is a need for an alternative strategy to determine the authenticity based on their botanical origins. Identification of honey origin based on specific chemical markers that guarantee the authenticity and ensure the traceability of honeys is an important and useful tool for quality assurance.

Considering the prospect of continuously increasing demands on SBH and the fast-growing bee-keeping industry, a robust and reliable method of determining the authenticity of honeys produced is necessary. To our knowledge, no study has yet been carried out using untargeted metabolites data to discriminate the SBHs according to their botanical origins. In addition, the relationship between botanical origins and antioxidant properties on SBH has yet to be established. In this study, the monofloral SBH with different botanical origins were investigated to determine the contribution of botanical origins to their antioxidant properties. First, the antioxidant properties of the monofloral SBH with different botanical origins were profiled. Next, the compositional differences between SBH were ascertained by a direct mass spectrometry (MS) approach coupled with chemometrics. Finally, an untargeted liquid chromatography-mass spectrometry (LC-MS) metabolomics approach was undertaken to identify the antioxidant compounds based on accurate masses, which could further explain the unique relationship between antioxidant and compositional profiles of monofloral SBH and their botanical origins.

## 2. Materials and Methods

### 2.1. Raw Honey Samples

Honey samples from stingless bees (*Heterotrigona itama*) were collected from multiple meliponid culture sites around Sabah, Malaysia, from September to December 2019. A total of seven meliponi culture sites with different predominant botanical origins were selected. The botanical floras studied are Mexican creeper (MC) (*Antigonon leptopus*), rubber (*Hevea brasiliensis*), dwarf mountain pine (DMP) (*Baeckea frutescens*), acacia (*Acacia mangium*), coconut (*Cocos nucifera*), starfruit (*Averrhoa carambola*) and agarwood (*Aquilaria malaccensis*). Five samples (~500 mL) from each flora type were obtained from commercial Meliponiculture farms. To collect a sample representing a flora origin, the honey pots were pierced with a clean, sharp wood stick, and the honey was aspirated from independent and cooperative honey pots with a portable suction pump. The samples obtained from a few beehives of the same flora type were pooled and labeled for its botanical origin. The collected honeys were kept in airtight plastic bottles and maintained in the dark at 4 °C during transportation. SBH samples were frozen at −20 °C until further analysis. 

### 2.2. Sample Extraction for Antioxidant Assays

SBH samples were extracted according to the method described by Borsato et al. [14]. Briefly, 50 g of honey sample was dissolved in 250 mL acidified water (pH 2 adjusted with concentrated hydrochloric acid) and homogenized with 100 g of polyaromatic adsorbent resin (Amberlite XAD-2, pore size 9 nm, particles size 0.3–1.2 nm) for 30 min using a magnetic stirrer. The mixture was transferred to a glass column (420 × 32 mm) and washed with acidified water (pH 2), followed by 300 mL of distilled water. The washed residue was eluted with 300 mL methanol. Excessive methanol was removed under reduced pressure at 40 °C. The concentrated extract was freeze-dried, then redissolved with methanol, and filtered with a 0.22 µm membrane filter prior to chromatographic and antioxidant analysis. 

### 2.3. Antioxidant Activities

#### 2.3.1. DPPH Free Radical-Scavenging Assay

The 1,1-diphenyl-2-picryl hydrazyl (DPPH) free radical scavenging activity of the honey extract was measured using a modified method described by Sousa et al. [27]. The freshly prepared 0.1 mM methanolic DPPH solution was mixed with 10 µL honey extract to a final volume of 200 µL in each working well of a 96-well plate and incubated for 30 min at room temperature. Absorbance was measured at 517 nm with Multiskan™ Sky microplate reader (Thermo Scientific, Waltham, MA, USA). The DPPH radical-scavenging activity of the honey extract was expressed in millimoles of Trolox equivalents per mg of extract (mM TE/mg).

#### 2.3.2. ABTS Free Radical-Scavenging Assay

The 2,2 azinobis 3-ethylbenzothiozoline-6-sulfonic acid (ABTS) radical scavenging activity of the honey extract was measured using a modified method from Silva et al. [25]. The ABTS radical cation was synthesized by reacting 7 mM aqueous ABTS solution with 2.45 mM potassium persulfate solution in the dark at room temperature for 16 h. Before analysis, the ABTS working solution was diluted with 70% ethanol to an absorbance of 0.700 ± 0.025 at 734 nm. In each well of a 96-well plate, 20 µL honey extract was diluted with ABTS working solution to a final volume of 200 µL and incubated for 10 min at room temperature. The absorbance was measured at 734 nm with the microplate reader. The scavenging capability of the ABTS radical of the honey extract was expressed in millimoles of Trolox equivalents per mg of extract (mM TE/mg).

#### 2.3.3. Ferric Reducing Antioxidant Power Assay

The ferric reducing antioxidant power (FRAP) activity of the honey extract was measured using a modified method by Tuksitha et al. [32]. The FRAP reagent was prepared by mixing 1 volume of 10 mM 2,4,6-tris-(2 pyridyl)-s-triazine (TPTZ) solution in 40 mM HCl, 1 volume of 20 mM ferric chloride solution and 10 volumes of 300 mM acetate buffer (pH 3.6). FRAP reagent was freshly prepared and kept at 37 °C in a water bath for at least 30 min. The reaction was performed by mixing 20 µL honey extract with 180 µL FRAP reagent in a 96-well plate and incubated for 10 min at 37 °C. The absorbance of the mixture was measured at 593 nm with the microplate reader. The reducing antioxidant power of the honey extract was expressed as millimoles of ferrous equivalents per mg of extract (mM Fe^2+^/mg).

#### 2.3.4. Oxygen Radical Absorbance Capacity Assay

The oxygen radical absorbance capacity (ORAC) of the honey extract was measured using the method described by Ranneh et al. [22]. Briefly, 150 µL 10 nM fluorescein (in pH 7.4, 75 mM phosphate buffer) and 25 µL honey extract were mixed in a 96-well plate well. After 10 min incubation at 37 °C, 25 µL of 153 mM 2,2-azobis(2-methylpropionamidine) dihydrochloride (APPH) solution was added to each well to initiate the reaction. Fluorescence reading was monitored using a fluorescent microplate reader at excitation/emission of 485/528 nm for 120 min with 60 s intervals. ORAC values of the honey extract were expressed as millimoles of Trolox equivalents per mg of extract (mM TE/mg).

### 2.4. Direct Mass Spectrometry Analysis of SBH Extracts

SBH was extracted by dissolving samples in water, followed by adding acetonitrile in a ratio of 1:1:1 (*w*/*v*/*v*). The mixture was vortexed and centrifuged at 10,000 rpm for 10 min. The top layer of acetonitrile was collected and sampled using a glass capillary (SG Lab, Singapore, Internal diameter 1.8 mm, external diameter 2.2 mm, length 100 mm) and introduced into RADIAN™ ASAP™ Ambient Mass Spectrometer (MS). The extraction steps were performed in 3 replicates for each sample. Each sample was introduced five times into the MS to gather a total of 15 data points for each honey sample. The data collected were analyzed using LiveID™ Software. Chemometrics analysis was performed with 10 possible PCA components, three linear discriminants, binning resolution of 1.0, and a mass range from *m/z* 100 to 800. 

### 2.5. Metabolite Profiling and Identification

Untargeted metabolite profiling of the methanolic honey extracts was performed with ACQUITY Ultra Performance LC™ (UPLC) I-Class System (Waters, Milford, MA, USA) coupled to Xevo™ G2-XS quadrupole-time of flight MS (Waters, Manchester, UK). Chromatographic separation was carried out in reversed phase (RP) and hydrophilic interaction chromatography (HILIC) separation. 

For RP separation, ACQUITY™ HSS T3 Column (100 mm × 2.1 mm, 1.8 µm) was used (Waters, Milford, MA, USA). Mobile phase eluent A was deionized water containing 0.1% formic acid, and eluent B was acetonitrile containing 0.1% formic acid. The gradient-elution was started at 0% eluent B to 30% for 10 min, 70% B for 5 min, 90% for 3 min, and back to 10% B in 2 min, making a total elution time of 20 min. The mobile phase flow rate and column temperature were maintained at 0.3 mL/min and 30 °C, respectively. For HILIC separation, ACQUITY BEH™ 100 mm × 2.1 mm, 1.8 µm Amide Column was used (Waters, Milford, MA, USA). Mobile phase eluent A was Acetonitrile, and eluent B was deionized water containing 10 mM ammonium acetate. Gradient elution was started at 0% B to 80% for 18 min and back to 0% B in 2 min, making a total elution time was 20 min. Mobile phase flow rate and column temperature were maintained at 0.4 mL/min and 45 °C, respectively. 

MS analysis was carried out in electrospray ionization (ESI) mode using negative and positive polarity for all samples with data independent MSE acquisition. MS parameters were as follows: *m/z* range of 50 to 1200 at 0.5 spectra/s, 2.0 kV capillary voltage, 6 eV collision energy for low energy function, and 20 to 40 eV collision energy ramp for high energy function, 120 °C source temperature, 600 °C desolvation temperature, 50 L/h cone gas flow and 1000 L/h desolvation gas flow. Raw data were imported into Progenesis™ QI Software (Waters, Manchester, UK) for peak picking, signal integration, normalization, and compound identification. Mass spectra libraries used in this study were the NIST MS/MS library and Waters METLIN MS/MS library. The list of candidates from library matches was filtered based on overall match score (≥30) and mass accuracy (≤10 ppm) to assign putative identity. Potential antioxidants were selected from the list of putatively identified metabolites based on the chemical classes. 

### 2.6. Statistical Analysis

Antioxidant results were presented as mean ± standard deviation. Statistical analysis was performed using XLSTAT statistical software (Free edition, 2016) (Addinsoft, rue Damrémont, Paris, France). Significant differences between means were determined by one-way analysis of variance (ANOVA) with post hoc Turkey’s test at a *p* < 0.05 confidence level. Metabolomics data was analyzed by MetaboAnalyst 5.0 for chemometric analyses [33]. Unsupervised principal component analysis (PCA) and supervised orthogonal partial least squares discriminant analysis (OPLS-DA) were used. R2 (cum) and Q2 (cum) were used to evaluate the fitness and predictive capability of the model. Variable importance of the projection (VIP) score was used to estimate the importance of each variable in PLS models. Outliers were determined using Hoteling’s T2 distribution with 95% and 99% confidence limits. Cross validation of the OPLS-DA models was performed using CV-ANOVA (*p* < 0.01) and permutation test (N = 100). Thereafter, S-plot was used to identify discriminant markers of each monofloral SBH, with cut-off values of *p* ≥ 0.5 and *p* (corr) ≥ 0.5, and further filtered using VIP ≥ 1.5. The relative fold-change analysis of discriminant markers of SBH was determined. Hierarchical clustering analysis (HCA) was used to illustrate the relative similarities and differences of metabolites among SBH.

## 3. Results and Discussion

### 3.1. Antioxidant Activities of Stingless Bee Honeys

The influence of botanical origins on the antioxidant activities of SBH was measured by single electron transfer (SET) antioxidant assays DPPH and ABTS, and hydrogen atom transfer (HAT) antioxidant assays, FRAP and ORAC (Table 1). DPPH measures the scavenging capability of antioxidants on DPPH free radicals [34], whereas FRAP assay measures the capability to convert the ferric (Fe^3+^) to ferrous ion (Fe^2+^) [29]. DPPH assays of SBHs ranged from 19.70 to 31.41 mM TE/mg and FRAP assays ranged from 45.5 to 89.3 mM Fe^2+^/mg (Table 1). The results are consistent with previous studies, which reported 4.3 to 23.6 mg ascorbic acid equivalent/100 g in SBH from 24 Melipona species and 0.8 to 28.2 mg ascorbic acid equivalent/100 g and FRAP values of 65.5 to 323.0 µM Fe^2+^/100 g for eight multifloral SBHs [27,35,36]. 

ABTS measures the decolorization of the pre-generated ABTS free radical in the presence of an antioxidant. In contrast, the ORAC assay measures the capacity of an antioxidant to capture the peroxyl radical produced by AAPH [22,23,24,25]. ABTS and ORAC values of the SBH ranged from 16.1 to 29.9 mM TE/mg and 69.0 to 167.6 mM TE/mg, respectively (Table 1). In the study by Ranneh et al. [22], the antioxidant activities of Tualang honey from *Apis dorsata* bees and Kelulut honey from Trigona bees were compared. ABTS and ORAC assays of Tualang SBH (176.7 and 29.6 µmol TE/g) were at least two-fold higher than *Apis dorsata* honeybee honey (45.9 and 22.3 µmol TE/g) at 25 uM honey concentration. Da Silva et al. [25] showed that the ABTS (EC50) values of methanolic extracts of nine multifloral stingless bee honey (*Melipona subnitida*) were varied from 21.2 to 53.1 µg/mL. In another study by Biluca et al. [32], the ORAC values of 13 multifloral stingless bee honeys ranged from 199 to 667 µM TE/100 g. 

Among the SBH of different botanical origins studied, acacia honey showed the highest antioxidant properties in ABTS assays (29.9 ± 1.7 mM TE/mg), FRAP assays (89.3 ± 2.4 mM Fe^2+^/mg), and ORAC assays (167.6 ± 2.1 mM TE/mg). In contrast, agarwood honey showed the weakest antioxidant properties in DPPH assays (19.7 ± 1.1 mM TE/mg), ABTS assays (16.0 ± 0.9 mM TE/mg), FRAP assays (45.5 ± 1.1 mM Fe^2+^/mg), and ORAC assays (77.0 ± 2.1 mM TE/mg) as summarized in Table 1. Some previous studies have also shown the high antioxidant potential of honeys from Acacia botanical origin from bees of different entomological origins [25,37,38]. In particular, Margaoan et al. [28] had shown a higher antioxidant potential of monofloral honey from Acacia, Manuka, and Clover botanical origin, increasing interest in monofloral honey for medicinal purposes.

### 3.2. Differentiation of Stingless Bee Honeys Using Direct MS and Chemometrics

Direct ambient MS coupled with chemometrics modeling has been applied in the analysis of many food products, including honey, syrups, Baijiu, cocoa butter, and herbs [27,28,29,30,32,33,34,35,36,39,40,41,42,43,44,45,46]. The screening approach allows rapid profiling of food products to determine if further analysis is required. To this end, the chemical fingerprint of SBH extracts from the panel of SBHs was obtained by a compact mass spectrometer with a dedicated atmospheric solids analysis probe (ASAP) and analyzed by chemometrics.

Principal component analysis (PCA) showed a total variance of 83.3% in positive polarity and 92.8% in negative polarity based on the first three principal components (PC). Supervised PCA-LDA analysis showed clear discrimination of the SBHs by botanical origins (Figure 1). In positive polarity, coconut honey was differentiated from the remaining honeys by PC1 (59.8%), agarwood honey was differentiated by PC2 (15.0%), and acacia honey was differentiated by PC3 (8.6%). In negative polarity, agarwood honey was differentiated by PC1 (73.3%), coconut honey was differentiated from the rest of the honeys by PC2 (12.7%), and acacia honey was differentiated by PC3 (6.8%). Figure 2 shows mass fingerprints from each loading plot, highlighting the masses contributing to respective PC. 

The clustering of SBH by botanical origins coincides with the level of antioxidant properties (Table 1). Acacia honey showed the highest values in ABTS assays (29.9 ± 1.7 mM TE/mg), FRAP assays (89.3 ± 2.4 mM Fe^2+^/mg), and ORAC assays (167.6 ± 2.1 mM TE/mg), and coconut honey showed highest values in DPPH scavenging activity (31.4 ± 0.5 mM TE/mg). In comparison, agarwood honey had relatively low antioxidant values for all the antioxidant assays evaluated. The chemometric models showed the greatest differentiation of coconut, acacia, and agarwood honeys from the other honeys, suggesting that the distinct differences in composition and antioxidant activities of SBH could be correlated. Attanzio et al. [47] reported that significant correlation between DPPH assays (r = 0.77) and FRAP assays (r = 0.88) with phenolic and flavonoid content in eight monofloral honeys of Sicilian black honeybee (*Apis mellifera* sp. *sicula*). Therefore, using metabolomics approaches, an investigation was carried out on the compositions of the SBH to identify antioxidant compounds uniquely associated with botanical origins. Since honey has complex chemical constituents with variations in physicochemical properties, multivariate statistics often facilitate the analysis to find any distinctive pattern based on honey origin. The integration of chemical analysis followed by chemometric analysis using various clustering and classification algorithms has been successfully used to identify honey origin [31].

### 3.3. Metabolites Profiling of Stingless Bee Honeys

Acacia honey was identified as a promising source of antioxidants compared to other honeys based on antioxidant assays. The distinct clustering based on mass spectral fingerprints has suggested unique differences in the chemical composition of acacia honey. An untargeted LC-MS metabolomics approach was used to obtain comprehensive polar and non-polar metabolite profiles of SBH to identify the possible antioxidant markers for acacia honey and its differences from other monofloral SBHs.

A total of 1893 features were matched against NIST MS/MS and Waters METLIN MS/MS libraries. The candidate compounds, which are related to antioxidant properties, include alkaloids, flavonoids (flavones, isoflavones, flavonols), phenolic acids (hydroxybenzoics, hydroxycinnamic), polyphenols, terpenoids (monoterpenoids, sesquiterpenoids) and vitamins were shortlisted [41] (Appendix A). Multivariate models were built using the shortlisted metabolite compounds, and 48.6% variability was obtained from unsupervised PCA based on the first two PC (Figure 3). The model showed excellent reliability with R2 (cum) and Q2 (cum) values of 0.96 and 0.93, respectively. The clustering obtained from the model has similar trends to the level of antioxidant properties, whereby acacia, coconut, and agarwood honeys which formed distinct clusters, have either the highest or lowest antioxidant properties among the honeys. Supervised OPLS-DA modeling and S-plots were performed, followed by a pairwise comparison between a single honey (acacia, agarwood, coconut, DMP, MC, rubber, and starfruit) and the remaining honeys as a group to identify key antioxidant markers of SBHs associated with botanical origins (Figure 4). A total of 47 antioxidant markers were identified to be enriched in respective botanical origins (Appendix A).

Hierarchical clustering (HCA), along with heat-map annotation of the relative abundances of antioxidant compounds, was used to demonstrate the clustering of SBH by similar metabolite profiles (Figure 5). As expected, the tightest clusters were formed by replicates of SBH from each botanical origin, and the unique antioxidant profile could be used to identify SBH by botanical origin. Two large clusters were formed in which DMP, MC, and rubber honeys were grouped in one cluster with a shorter distance between nodes of the dendrogram, indicating high similarity among the honeys. Agarwood, coconut, acacia, and starfruit honeys were grouped in another cluster with larger distances between nodes of the dendrogram, indicating low similarity among the honeys. The similarity of metabolite profiles of DMP, MC, and rubber honeys is the most probable reason for the similar antioxidant properties (Table 1).

Acacia honey showed the highest antioxidant properties among the SBH tested and was enriched with polyphenols, flavonoids, and alkaloids based on metabolomics data. These putatively identified compounds are 2-butyl-3-phenyl-2-propen-1-al, tamarixetin 3-glucosyl-(1->2)-galactoside, 2-indolecarboxylic acid, 6-methoxyluteolin, isorhamnetin 3-galactosyl-(1->4)-rhamnosyl-(1->6)-galactoside, and matairesinol (Appendix A). The enrichment in flavonoids and their derivatives by three to five-fold compared to other SBH explains the higher antioxidant properties in acacia honey. These results support previous studies by Suarez et al. [48], which reported the isorhamnetin containing fraction in SBH from *Tetragonula biroi* had antibiotic properties against multidrug-resistant *Staphylococcus aureus.* Sousa et al. [27] reported that polyphenols level corresponds to the antioxidant activities of SBH. These results show that the botanical origin influences the antioxidant and medicinal potential of SBH, leading to differences in quality.

## 4. Conclusions

This study shows that the antioxidant properties of SBHs are related to the unique antioxidant metabolites of foraged nectar from specific botanical origins, and these are reflected in mass spectral fingerprints from ambient mass spectrometry and metabolomic profiling. These antioxidant metabolites consist of alkaloids and flavonoids, which are known to be strong antioxidants. Acacia honey showed the highest antioxidant properties and distinct antioxidant profile and, therefore, could have better medicinal properties compared to SBH from other botanical origins. Therefore, to meet the growing demand for medicinal SBH, stingless bee farming can focus on specific botanical origins that confer higher antioxidants. This study revealed an untargeted metabolomics approach is a powerful tool with wide coverage, high throughput, and strong robustness in discriminating the SBHs according to their botanical origins. A holistic approach by associating metabolite data with supervised chemometric analysis could lead to identifying specific compound markers related to a botanical origin that may serve for authentication purposes. Hence, this approach could be useful for the authority to guarantee honey authenticity and traceability and increase consumer confidence in locally produced honeys. However, more research is needed to verify the selected compounds as reliable markers for quality control.

## Figures and Tables

**Figure 1 foods-12-02404-f001:**
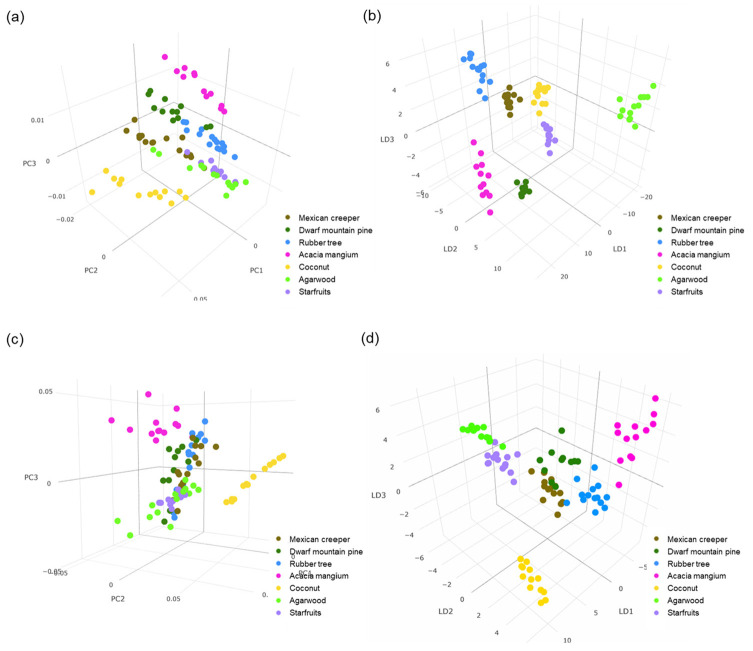
Unsupervised PCA plots of stingless bee honeys from different botanical sources in (**a**) positive; and (**b**) negative ionization modes. Supervised PCA-LDA plots of stingless bee honeys in (**c**) positive ionization; and (**d**) negative ionization modes.

**Figure 2 foods-12-02404-f002:**
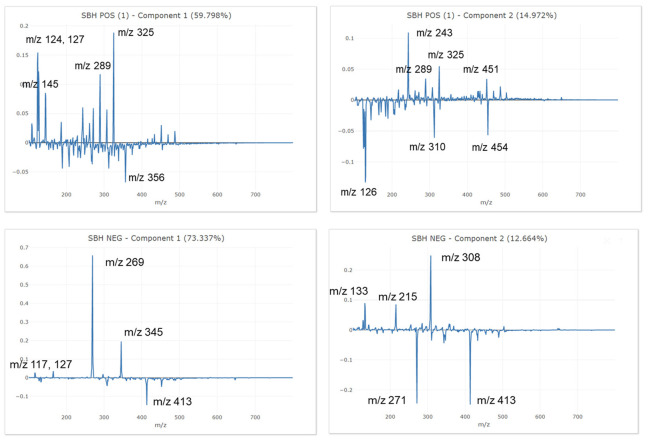
Mass fingerprints obtained from each component of principal components analysis of stingless bee honeys from different botanical origins.

**Figure 3 foods-12-02404-f003:**
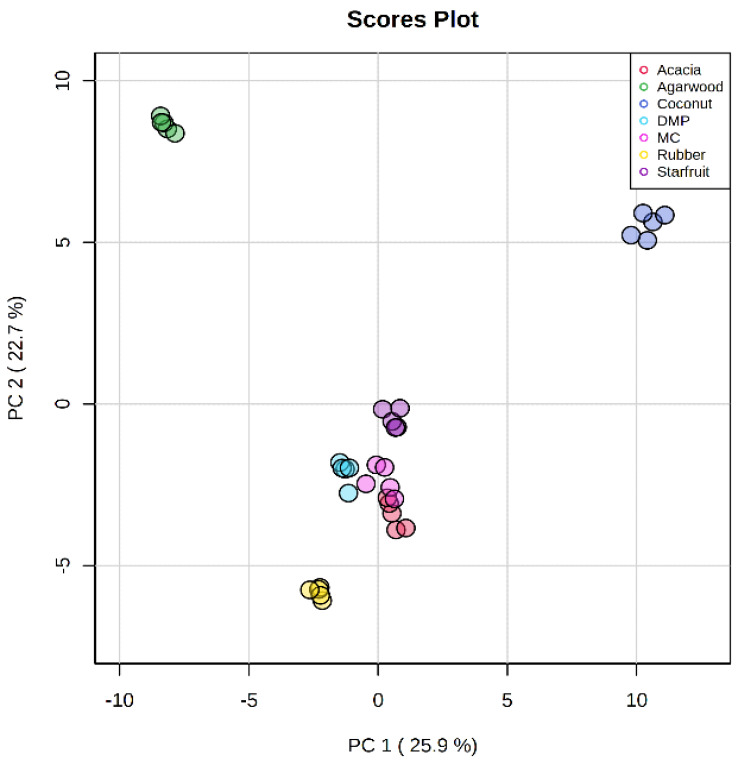
PCA score plot of stingless bee honeys from different botanical origins constructed based on 105 identified antioxidative compounds.

**Figure 4 foods-12-02404-f004:**
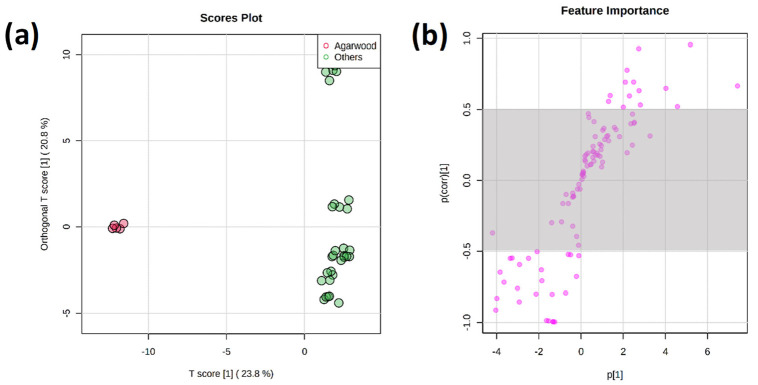
A representative (**a**) OPLS-DA score plot; and (**b**) S-plot of agarwood stingless bee honey against remaining stingless bee honeys (acacia, coconut, DMP, MC, rubber, starfruit) based on the 105 identified antioxidative compounds. The cutoff values of *p* ≥ 0.5 and *p* (corr) ≥ 0.5 were used for the selection of candidate marker compounds from the S-plot.

**Figure 5 foods-12-02404-f005:**
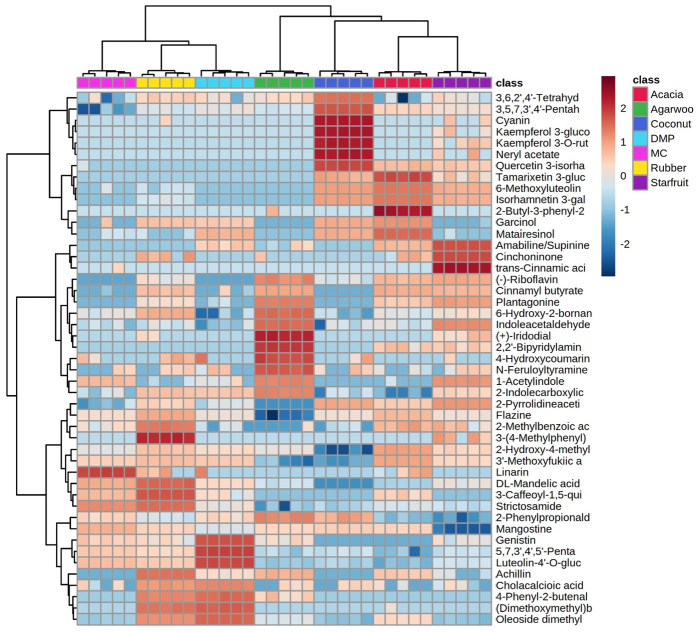
HCA dendrogram with heat map visualization based on 47 discriminant antioxidative compounds from stingless bee honeys from different botanical origins. Red indicates higher metabolite abundance compared to the mean, whereas blue indicates lower metabolite abundance compared to the mean.

**Table 1 foods-12-02404-t001:** Antioxidant activities of stingless bee honeys from different botanical origins.

Honey	DPPH (mM TE/mg)	ABTS (mM TE/mg)	FRAP (mM Fe^2+^/mg)	ORAC (mM TE/mg)
Acacia	24.42 ± 1.91 ^bc^	29.85 ± 1.68 ^a^	89.27 ± 2.44 ^a^	167.55 ± 2.14 ^a^
Agarwood	19.70 ± 1.08 ^e^	16.05 ± 0.86 ^d^	45.50 ± 1.08 ^e^	77.25 ± 2.93 ^e^
Coconut	31.41 ± 0.46 ^a^	22.23 ± 1.71 ^c^	66.94 ± 1.75 ^c^	98.60 ± 3.63 ^c^
DMP	21.15 ± 1.07 ^de^	20.57 ± 1.07 ^c^	56.84 ± 1.91 ^d^	69.03 ± 3.71 ^f^
MC	23.28 ± 1.59 ^cd^	27.63 ± 1.22 ^ab^	66.89 ± 1.12 ^c^	87.59 ± 3.18 ^d^
Rubber	26.65 ± 0.74 ^b^	25.35 ± 0.75 ^b^	73.61 ± 3.29 ^b^	123.79 ± 1.04 ^b^
Starfruit	19.96 ± 1.81 ^e^	17.45 ± 1.04 ^d^	55.92 ± 1.92 ^d^	91.70 ± 4.00 ^d^

Values are reported as mean ± standard deviation (*n* = 5). Different superscript letters indicate a significant difference in *p* < 0.05. TE: Trolox equivalent; MC: Mexican creeper; DMP: Dwarf mountain pine.

## Data Availability

The datasets used and/or analyzed during the current study are available from the corresponding author upon reasonable request.

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
