# Peer review of "Antioxidants Discovery for Differentiation of Monofloral Stingless Bee Honeys Using Ambient Mass Spectrometry and Metabolomics Approaches"

_foods, 2023, doi:10.3390/foods12122404_

Round 1

Reviewer 1 Report

The authors reported “Antioxidants Discovery for Differentiation of Monofloral Stingless Bee Honeys using Ambient Mass Spectrometry and Metabolomics Approaches”. The paper is very well written and contributes Anun-targeted liquid chromatography-mass spectrometry (LC-MS) metabolomics approach was undertaken to identify the antioxidant compounds that could explain the unique antioxidant and compositional profiles of monofloral SBH by botanical origin. scheme for high feasibility, which enables higher antioxidant and medicinal properties. Overall recommendation: “major revision”.

For the benefit of the reader, however, a number of points need clarifying and certain statements require further justification. Some detailed information is provided following. 

Specific comments: 

1.     The topic is novel but the application proposed is a little poor.

2.     The author needs to consider the advantages of untargeted liquid chromatography-mass spectrometry deeply.

3.     Please state more information about the application status and development prospects of SBH.

4.     The study lacks a sufficient explanation of the simulation results.

5.     The study needs to discuss more deeply, and highlight important findings.

6.     The significance of this paper is not expounded sufficiently. The author needs to highlight the innovative contributions.

7.     In page 7, the figures are a bit blurry. Please consider replacing them with clearer ones.

8.     There was little explanation of the fundamentals of the study design.

9.     There is no experimental comparison of the algorithm with the previous study, please add it.

Author Response

On behalf of the authors, I would like to thank the reviewer for the valuable comments. The manuscript has been revised accordingly based on the comments of the reviewers. The detailed responses to the comments (point-to-point) could be found in the attached file.  Please see the attachment. Thank you.

Reviewer 2 Report

Manuscript ID: foods-2236682

In the submitted manuscript, author investigated the antioxidant property of stingless bee honey (SBH) from different botanical origins. And direct ambient MS and HRMS were performed for finding possible discrimination markers. Author aimed to discovery the antioxidant marker for SBH, however, the correlation between the proposed markers and the antioxidant activity had not been provided and demonstrated. In my opinion, the three parts in this study were independent and short of systematic design.

Some other comments:

1. line 82, section 2.1: how many samples were used for the antioxidant assay and the chemometric study?

2. line 145, section 2.4: I suggest the use of the same methanolic exact in section 2.3 and 2.5 for ambient MS study, otherwise the analysis of antioxidant compounds would be meaningless.

3. line 235, n=5, it means five samples or just five replicates from one sample?

4. line 261, (b) should be (c).

5. line 321, figure 6, compounds should be listed in the y-axis.

Author Response

(The authors gave the same response as above.)

Reviewer 3 Report

line 68-69 – “The relationship between botanical origins and antioxidant properties on SBH has 68 hitherto not been established based on botanical diversity in Sabah.” –please replace the word hithero with an more commonly used word. What is Sabah?

There is a lack of a clearly defined research objective/aim in the last paragraph of the introduction. The authors described broadly only the scope of the study.

Latin names of bees or plants need to be given in italics, please check the entire text as authors do not always use such notation.

line 85- what is meliponi?

Raw Honey samples- How much honey of each type was taken and stored?

line 102-103- How much methanol was used to dissolve the lyophilizate? Was the filtered extract used only for chromatographic analyses or also for DPPH, ABTS, FRAP and ORAC analyses? please specify

Figure 2 - All 4 PCA graphics need resolution improvement because the legends of the colored dots are very faintly visible and thus the graphics are not very informative. Where is description of graph c?

Figure 3- please verify the title of the graphs as they are more like mass spectra of compounds.

Figure 4 - The same situation as for Figure 2.

Results- In the results section, authors report and describe only their results. Only in the discussion section do they compare their results with others in the literature and discuss us observations. Therefore, comparisons with results from the literature should be moved to the discussion section.

I propose to combine text of the “results” and “discussion” chapters and create a “results and discussion” chapter that is allowed in the journal. This will make the whole thing more coherent and have a better tone.

In the current situation, the discussion of results is very sparse and should be expanded especially in view of metabolomic and chemometric analysis.

I have no comments on the conclusions. They result from the authors' research.

Author Response

(The authors gave the same response as above.)

Round 2

Reviewer 1 Report

The study could be accepted in present form.

Author Response

Thank you for your recommendation. We have revised the sampling to clarify the concern of the academic editor. With this revision, hopefully the research design has been improved.  

Reviewer 3 Report

The authors addressed all comments and remarks. After reading the article after the corrections made, I believe that the work has been significantly improved in terms of quality and content conveyed.

Author Response

Thank you for the recommendation.